# Capacity Building Enables Unique Near-Fault Observations of the destructive 2025 $M_w$ 7.7 Myanmar Earthquake

Ssu-Ting Lai[1], Kyaw Moe Oo[2], Yin Myo Min Htwe[2], Tin Yi[2], Htay Htay Than[2], Oo Than[2], Zaw Min[2], Tun Minn Oo[2], Phyo Maung Maung[3], Dino Bindi[1], Fabrice Cotton[1,4], Peter L. Evans[1], Andres Heinloo[1], Laura Hillmann[1], Joachim Saul[1], Christoph Sens-Schönfelder[1], Angelo Strollo[1], Frederik Tilmann[1,5], Graeme Weatherill[1], Ming-Hsuan Yen[1,4], Riccardo Zaccarelli[1], Thomas Zieke[1], and Claus Milkereit[1]

[1]GFZ Helmholtz Centre for Geosciences, Potsdam, Germany
[2]Department of Meteorology and Hydrology (DMH), Naypyitaw, Myanmar
[3]Nanyang Technological University (NTU), Singapore
[4]Institute of Geosciences, University of Potsdam, Potsdam, Germany
[5]Institute for Geological Sciences, Freie Universität Berlin, Germany
**Correspondence:** Dino Bindi (dino.bindi@gfz.de)

**Abstract.** We present an overview of Naypyitaw, Myanmar station (NPW), managed by the Department of Meteorology and Hydrology (DMH) and part of the GEOFON (GE) network. the station is equipped with both strong-motion and broadband sensors and is situated 2.6 km from the Sagaing Fault, providing an exceptional near-fault recording of the $M_w$ 7.7 earthquake that occurred on March 28, 2025. The installation and ongoing maintenance of NPW are the result of a collaborative effort between DMH and the GFZ Helmholtz Center for Geosciences (GFZ) prompted by the GFZ International Training Course on seismology and seismic hazard assessment (ITC) in 2016. In this study, we provide background information on the collaborative effort that led to the installation of the only station providing near-fault, on-scale measurements of the 2025, Myanmar earthquake. Given the widespread interest for data recorded by station, we describe the instrumental settings in detail, and how to access data and metadata for station NPW. Given the relevance of the near-fault recordings at NPW not only for constraining the rupture process of the mainshock but also for engineering seismology applications, we analyze key features of the mainshock from an engineering seismology perspective. This includes an examination of ground motion amplitudes, frequency content, and response spectra, and near-fault effects such as fling effect and pulse-like motion. The high-quality near field data at NPW provide valuable information for seismic hazard assessment in the region and offer useful constraints for studies investigating the rupture characteristics of the mainshock, which preliminary findings suggest to have propagated at supershear speed.

# 1  Introduction

On March 28, 2025, a moment magnitude $M_w$ 7.7 earthquake (Quinteros et al., 2021) struck central Myanmar, exposing the second and third largest cities of Myanmar, Mandalay and Naypyitaw, respectively, to intensity IX shaking (U.S. Geological Survey, 2025c) and causing extensive damage and more than 3000 fatalities (U.S. Geological Survey, 2025b). The event occurred along the Sagaing Fault, a major right-lateral strike-slip fault that slips at $\sim$2 cm/yr and accommodates about half of the transverse motion between the Indian and Sunda plates (Socquet et al., 2006). Myanmar is a seismically active region with high seismic hazard (Yang et al., 2023) and specifically the fault segment between Mandalay and Naypyitaw had been identified as a seismic gap (Hurukawa and Maung Maung, 2011). Its current national seismic monitoring network (MM), which is operated by the Department of Meteorology and Hydrology (DMH) of Myanmar, has been initiated in 2016 with the installation of 5 seismic stations with support of the US Geological Survey and USAID (Thiam et al., 2017). With the support of the Indian National Centre for Ocean Information Services (INCOIS) and Regional Integrated Multi-hazard Early Warning System (RIMES, based in Thailand), an additional 4 stations were added (Figure 1). For several years, the Earth Observatory Singapore operated a permanent broadband network of 30 stations (Wu et al., 2021, e.g.), which completed in 2024. Additionally, several temporary networks of shorter recording times of 1-2 years provide information of the seismic structure (e.g. Tilmann et al., 2021; Sandvol et al., 2018; Bai et al., 2020) In spite of these efforts, Myanmar's permanent seismological infrastructure remains sparse, and all networks suffered station outages due to travel restrictions caused by Covid and the political situation.

As part of a long-term collaboration between the GFZ German Research Centre for Geosciences[1] (GFZ) and the DMH, an effort that also includes capacity building, the NPW station was installed in Naypyitaw and integrated into the global GEOFON Seismic Network (GE) (GEOFON Data Centre, 1993). Station NPW provides the only near-fault, on-scale recordings of the mainshock and several large aftershocks, offering a unique dataset to constrain the source rupture characteristics. Stations NGU, YGN, and KTN of the MM network also provided strong motion recordings of the mainshock but at much larger distance to the fault (Figure 1). NPW data also enable the analysis of near-fault ground motion parameters that are critical from an engineering seismology perspective. Furthermore, preliminary analysis of the Myanmar

---

[1]Current name: GFZ Helmholtz Centre for Geosciences

earthquake indicates that its rupture propagated at supershear velocities (Vera et al., 2025); these data thus provide a rare on-scale near-fault record during the supershear phase of the rupture.

In this study, we start by outlining the ongoing collaborations that enabled the installation of station NPW. We then provide detailed information on the station's metadata, data availability, and instrumental configuration, including a discussion of issues encountered in the immediate aftermath of the mainshock and the corresponding solutions. Finally, we present an engineering seismology analysis of the mainshock recordings, focusing on key parameters such as peak ground motion values, response spectra, spectral characteristics, and near-fault effects, specifically the identification of fling step and pulse-like motions. The analysis of near-source recordings highlights the uniqueness of the data set constructed with the recordings of station NPW.

## 2 Background collaboration

For more than a decade, the GFZ and the DMH of Myanmar have been developing a long-term partnership to strengthen seismic monitoring, hazard assessment and scientific capacity building in Myanmar. The first contacts were established in 2010 with initial cooperation within joint regional seismological initiatives. In 2016, GFZ and DMH jointly organized the International Training Course on seismology and seismic hazard assessment (ITC) in Naypyitaw (https://www.gfz.de/fileadmin/gfz/sec21/pdf/ITKurse/2016M yanmar/web_programme.pdf); also see (Milkereit et al., 2023). During the training course, scientists from Myanmar and the wider Southeast-Asia region were trained in Seismology, Seismic Data Analysis, Hazard Assessment and Risk Mitigation. For the first time, installation of a permanent seismic station (NPW) and a SeisComP system for acquisition and analysis (Helmholtz Centre Potsdam GFZ German Research Centre for Geosciences and GEMPA GmbH, 2008) were added as additional components to the training program and handed over to the local hosting institution at the end of the training (Figure 2). The standard programme of the ITC was thus complemented with transfer not only of hardware, but more of additional technical know-how, building on the lessons learned from establishing and maintaining the GEOFON network (Quinteros et al., 2021; GEOFON Data Centre, 1993). During the ITC, general introductory lectures on SeisComP introduced all participants to the basic use and focused on how to obtain relevant metadata and waveform data from other data archives operated by EIDA (Strollo et al., 2021) and EarthScope IRIS; the participants also practised interactive event location using the SeisComP

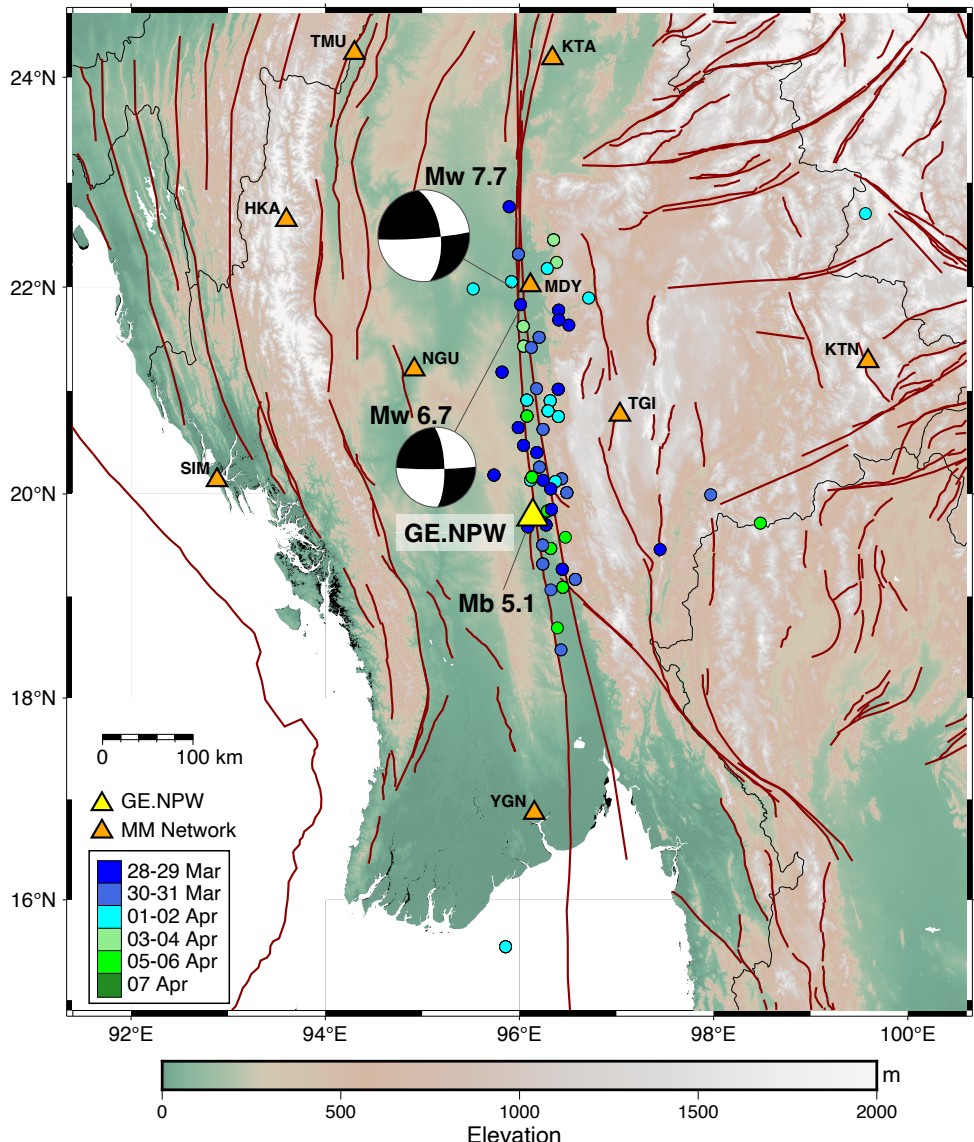

**Figure 1.** The mainshock and 61 selected earthquakes with magnitudes greater than 3 recorded up to 7 April 2025. The focal mechanisms of the $M_w$ 7.7 mainshock and the $M_w$ 6.7 aftershock are obtained from the Global Centroid Moment Tensor Project (https://www.global cmt.org/). The triangles indicate the locations of the GEOFON station NPW (yellow) and of the Myanmar National Seismic Network (MM) stations (orange). Active faults (shown in dark red) are sourced from the GEM Global Active Faults Database (Styron and Pagani, 2020). Map is produced with Generic Mapping Tool software (Wessel et al., 2019), with topography generated using the ETOPO1 global relief model (Amante and Eakins, 2009).

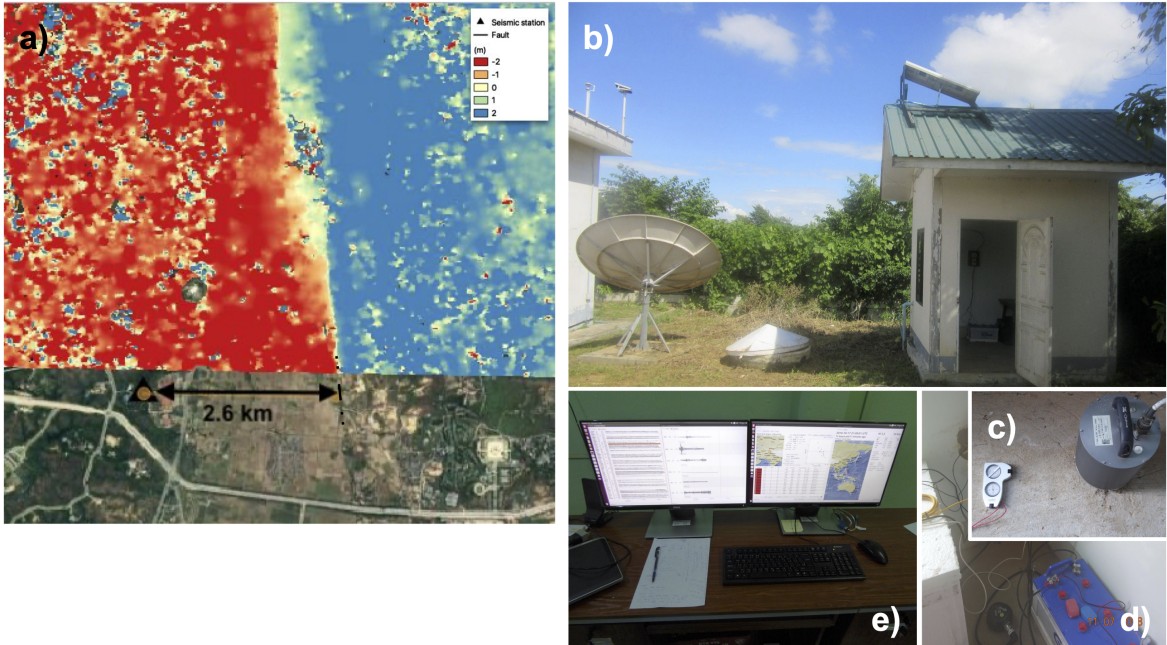

**Figure 2.** NPW station at DMH: a) location of station NPW with respect to the main rupture plane of the 2025, Mw 7.7 Myanmar earthquake as depicted by pixel offset tracking (Strozzi et al., 2002) on Sentinel 1 (https://sentiwiki.copernicus.eu/web/s1-mission) and Sentinel 2 (https://sentiwiki.copernicus.eu/web/sentinel-2) data (courtesy of M. Motagh and the Remote Sensing for Geohazards group, GFZ Helmholtz Centre for Geosciences; see Vera et al. (2025) for details; Map data ©2025 Google); b) view of the shelter housing the NPW instruments; c) broadband (before being covered by thermal insulation) and d) accelerometer; fixed to the floor of the shelter without thermal insulation; e) the SeisComP system used during the 2016 training course, integrating all regional stations and a subset of global stations.

graphic user interface with simulated playbacks. The later part, dedicated to DMH staff only, went into more detail on the practicalities of installing SeisComP and integrating into the real-time workflow the new seismic station NPW at DMH, other Myanmar National Seismic Network (FDSN network code MM)

stations (Department of Meteorology and Hydrology - National Earthquake Data Center, 2016), which are acquired from a separate local acquisition server, as well as stations from other networks in Southeast Asia and the rest of the world. The availability of high-quality strong-motion recordings, particularly from near-fault stations such as NPW, provides valuable data for advancing our understanding of seismic hazard and for informing future engineering design and mitigation efforts in the region.

As a further outcome of the deeper cooperation fostered during the workshop, a temporary seismic array consisting of 30 broadband stations was deployed in northern Myanmar (Tilmann et al., 2021), in close

cooperation with other international activities from the US (Sandvol et al., 2018) and Singapore (Wang et al., 2019), and with consideration of the temporary Chinese experiments conducted in Myanmar shortly before (Mon et al., 2020). During the preparation and deployment of the array, additional spare hardware was handed over to DMH, and maintenance and capacity-building activities were also carried out at the NPW co-operated station.

## 3 Instrumental settings

The DMH/GEOFON seismic station in Naypyitaw (NPW) is the first station in the GE network that was established during an ITC event. The operation of this station, like that of nearly all the other GE stations, is based on a clear commitment from the partners, in this case, DMH, to commit to long-term joint operation and ownership by providing local facilities and technical expertise in case of maintenance actions being needed. In return, GEOFON provides hardware as needed, capacity building (in this case embedded in the ITC), data curation and preservation, and a long-term commitment to joint operations, which includes remote support during maintenance actions, if required, and the shipping of replacement parts that cannot be obtained locally. The NPW station is located in Naypyitaw, Myanmar, within the DMH facilities at 19.78°N and 96.14°E, at an elevation of 158 m. It is only 2.6 km from the rupture plane and 246 km from the epicentre of the 2025, $M_w7.7$ mainshock (Figure 2a).

To prepare for the installation, DMH provided a shelter (Figure 2b) with a 2 m deep vault adjacent to it. They further contributed an existing Güralp CMG-3ESPC 30 sec sensor, a continuous power supply and connectivity via the local network of the seismology department building. During the preparatory work, which took place in early 2016, GFZ provided additional Güralp hardware, including a Fortis accelerometer, a DM24 digitizer and an EAM local computer unit, which enables direct seedlink streaming (Figure 2). In addition to the seismological hardware, GFZ provided a mobile router for backup communication via a protected Virtual Private Network (VPN) link and a power control module. The station is powered by the local mains power with an additional under-voltage protected battery as buffer, to keep the station running for a few days in case there would be a power outage. As the vault was flooded just before installation, the broadband sensor, the accelerometer, and all the other hardware were housed in the shelter (Figures 2b,c). The mode of the power spectral density (Figure 3) falls within the lower and upper limits of the New Noise Model proposed by Peterson (1993). In particular, it is less than 10-20 dB

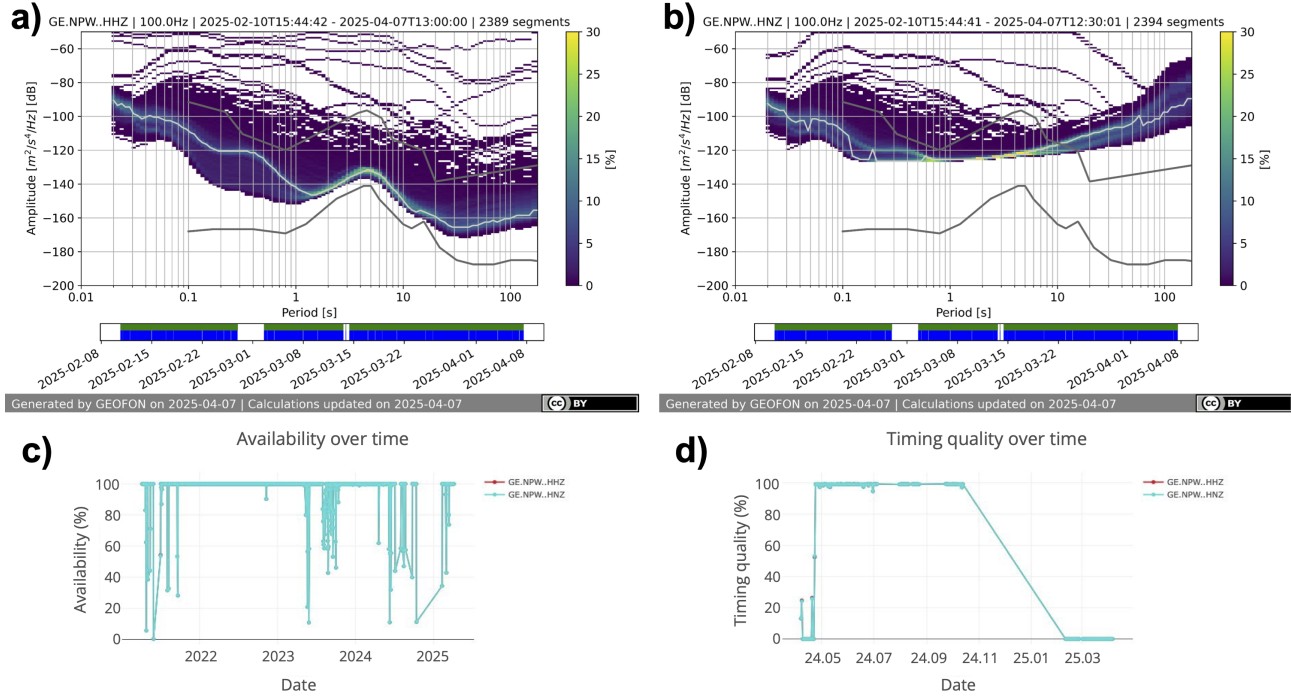

**Figure 3.** Power Spectral Density (PSD) of the vertical components of both sensors for the data available in 2025; note that the colour scale was reversed compared to ObsPy default in order to enhance visibility of the mode line, a) broadband sensor, b) strong-motion sensor (see Data Availability section for links to corresponding plots for all components and user-defined time spans); c) daily availability for the last 4 years (since 01.01.2021), d) timing quality as stored in the file headers for one year time span.

from the lower limit for periods longer than 1 second. Since the sensor showed reasonable noise levels for the wide band sensor, it was decided to leave the sensors in the shelter rather than move them to the vault and risk damage from repeated floodings. In terms of noise level (Figure 3), the station is well suited for local, regional and teleseismic monitoring, despite its location in an area of urban activity. The broadband sensor allows clear identification of the primary and secondary microseismic peaks. The increasing local anthropic noise is clearly visible for periods shorter than 1 s. The strong-motion sensor performed as expected for periods shorter than 1 s, enabling the dynamic range extension to record large near-fault events up to $2g$, which is the full scale to which it is set.

Therefore, although NPW is not installed in a remote and quite area, it can be considered a good-quality station for intermediate and long periods. In our view, it represents a good compromise between quality and availability in a region with sparse coverage; the setup was discussed and agreed upon with

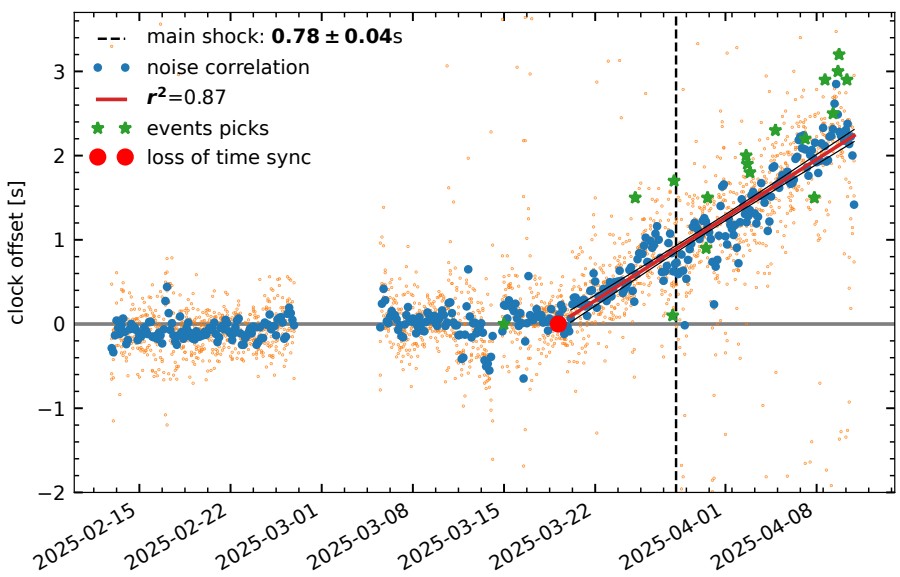

**Figure 4.** Timing of NPW recordings from earthquake picks and ambient noise correlation. Stable timing is confirmed until March 19th (red dot) when the internal clock starts to drift with about 0.1s per day. Orange dots show independent estimates of ambient noise correlations from the north and vertical components with all components of station MM.NGU (Sens-Schönfelder et al., 2014). Blue dots represent the joint estimate from all components. Stars indicate the clock offset inferred from earthquake arrival time picks. Red line with black confidence limits indicates the least squares fit of the clock drift. The main shock is indicated by the vertical line.

the local partners and adapted to the actual capacity and needs of long-term operation. In fact, since its installation, the station has been operating with several intervals where the data flow was interrupted, but always restored by DMH staff with only remote support from the GEOFON maintenance team. The overall data availability since installation is $\sim 80\%$. Figure 3c shows the availability for the last 4 years (since 01.01.2021) with a large gap between 13 October 2024 and 10 February 2025 with an overall average availability of 82% for the given period (90% since January 2025 and 100% since 28/03/2025 (day of the mainshock) to 07/04/2025). The station's timing quality has been unstable since installation, however, despite attempts to change parts of the GPS hardware. For users of these data, we further note that the timing quality reported by the data logger seems to be consistent with the time since last lock until the beginning of the large gap in October 2024 (Figure 3d), but when data flow was restored in February 2025, the timing quality was no longer consistently reported, and can no longer be considered

a reliable indicator. Nevertheless, sporadic GPS locks occurred every few days, with the last known GPS lock before the mainshock on 23 March 2025. Triggered by the observations of growing residuals in arrival times we investigated the timing of NPW using ambient seismic noise cross-correlation (Sens-Schönfelder, 2008) of the HH channels with the neighboring stations MM.NGU, MM.TGI, and TM.MHIT (Figure 1). Correlations of NPW's HHZ and HHN channels with NGU provide most stable estimates of the clock offset, as illustrated in Figure 4. We find that NPW lost GPS synchronization on March 19 at 4am ($\pm18$h) when the clock starts to drift linearly with a rate of $1.14 \times 10^{-6} \pm 6.6 \times 10^{-8}$, i.e. $98.6 \pm 5.7$ms/d (regression coefficient $r^2 = 0.87$). At the time of the mainshock, the accumulated clock error was $775 \pm 40$ ms. Observations with other stations confirm these findings, however, with less accuracy.

The NPW station plays a crucial role in regional and global seismic monitoring, providing valuable data for seismological research and contributing to earthquake hazard studies by improving the completeness of earthquake catalogs. The station is in the International Registry of Seismograph Stations and has contributed to more than 3000 P-phase observations at the ISC (International Seismological Centre , 2025), see https://www.isc.ac.uk/cgi-bin/stations?stacode=NPW.

Following the $M_W$ 7.7 earthquake on March 28, 2025, the contact was initially lost a few seconds after the P onset. The condition of the station was unknown to GEOFON staff at that time. The station started transmitting data again on April 1, 2025, four days after the earthquake. By remote connection, it was confirmed that the station had continued recording throughout, and just data transmission was lost during the data gap, such that it was possible to backfill the data holdings. Given the amount of damage incurred to the power and communications infrastructure, the continuous operation of the station and quick restoration of connectivity are remarkable.

Given low prior usage of the strong motion data, GEOFON staff identified a serious metadata issue affecting three strong motion components; in the wake of clarifying this issue, a more subtle issue was also identified with the metadata of the broadband instruments. As a result, a public warning to data users was put out immediately (https://geofon.gfz.de/forum/t/metadata-for-ge-npw/32635). As soon as the open questions could be clarified, which needed checking with the manufacturer, the metadata were updated and a notification was put out to warn users to re-download the metadata.

The station proved to be instrumental in recording this seismic event, providing critical data for the rapid understanding of some basic characteristics of the earthquake, as presented in the following sections of this article.

## 4 Data quality and parameters of engineering interest

In order to evaluate the quality of the data and metadata of station NPW and to characterize the recordings of the 2025 sequence from an engineering seismology perspective, we use the stream2segment software (Zaccarelli et al., 2019) to download the seismic waveform data from the GEOFON node (https://geofon.gfz.de/fdsnws/dataselect/1/) of EIDA (https://www.orfeus-eu.org/data/eida/) and populate a local PostgreSQL database. Station metadata are retrieved from GEOFON station service (https://geofon.gfz.de/fdsnws/station/1/). Event metadata are retrieved from the EMSC event web service (http://www.seismicportal.eu/fdsnws/event/1/) to compile a catalog of earthquakes with magnitude greater than 3, occurring since 2021, within a radius of 5.5° from station NPW. From the continuous waveform data, we extract 5-minute windows centered on the theoretical P arrival of each event computed considering the AK135 global velocity model (Kennet et al., 1995).

To assess the quality of the data and evaluate their potential for both seismological applications, such as constraining the spatial and temporal evolution of the rupture process, and engineering seismology studies, we conducted a series of analyses based on the mainshock recordings:

- **Broad band displacement and static offset**. The NPW recording of the $M_w$ 7.7 mainshock provides the rare opportunity to investigate near-fault ground motion data. Figure 5 shows the static displacement obtained through double integration of the unfiltered, instrument-corrected acceleration. Before each integration, a linear trend was removed from the entire time window. The trend was based on straight-line fitting of 5 minutes of noise immediately preceding the P onset. To determine the static offset and to flatten the coda trace, a simple quadratic function was fit to first 2 minute of the double-integrated data after the displacement stabilized. The linear and quadratic terms were then removed from the data. The retrieved static displacement on the north-south (NS) component, which is almost parallel to the fault strike, is 1.6 m, whereas on the east-west (EW), it is about -0.13 m. The static displacement offset measured along the fault parallel component (i.e., the NS one), which corresponds to unidirectional long-period velocity pulse, is known in the engineering seismology context as fling-step, or fling velocity-pulse (e.g., Hisada and Tanaka, 2021).

For the following analysis each trace is detrended and 5% tapered at both ends before being band-pass filtered using a zero-phase (acausal), fourth-order Butterworth filter. For the mainshock, the high-pass corner frequency is fixed at 0.02 Hz, while for aftershocks, it is adapted based on magnitude (Puglia

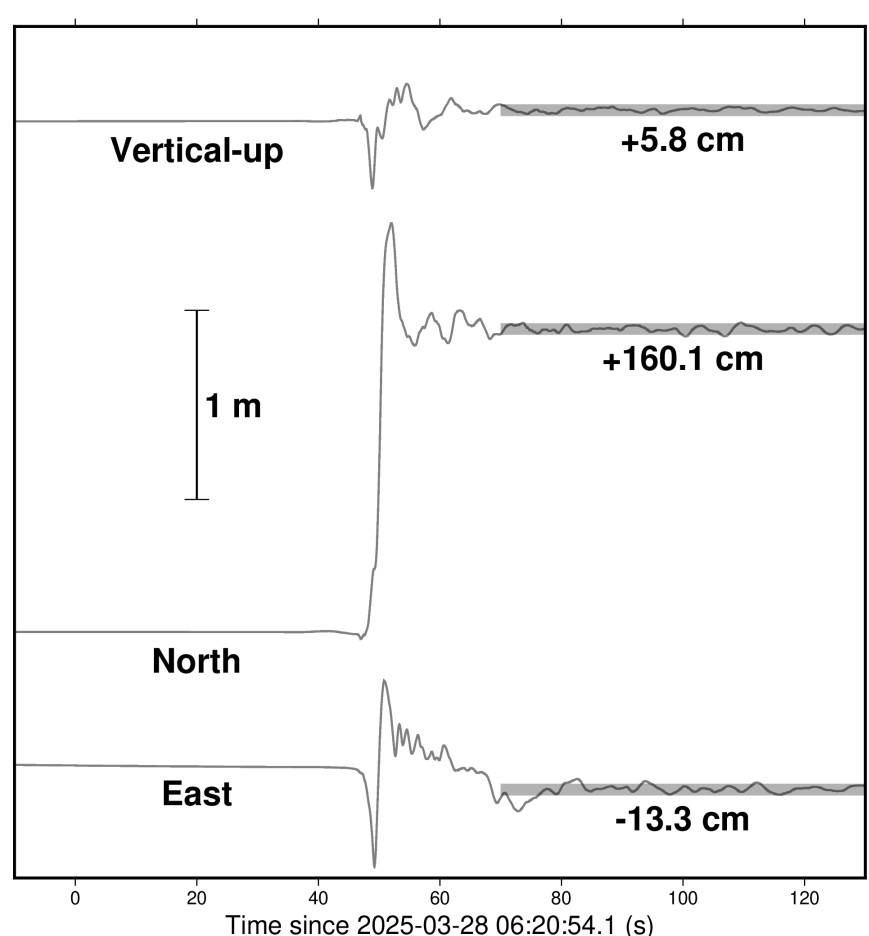

**Figure 5.** Displacement time histories of the $M_w$ 7.7 mainshock recorded at station NPW, obtained through double integration and baseline corrections of the instrument corrected acceleration. The timing correction of -0.78 s was applied to the data already.

et al., 2018). The low-pass corner frequency is set to 40 Hz. The instrumental response is removed, and the resulting acceleration time series are integrated to obtain velocity. Finally, acceleration and velocity response spectra are computed assuming 5% critical damping.

- **Peak ground acceleration and velocity**. To analyze the earthquake recordings from an engineering seismology perspective, Figure 6 presents the high-pass filtered acceleration and velocity time histories. The peak ground accelerations (PGAs) for the vertical (Z), NS (fault-parallel), and EW (fault-normal) components are 10.53, 6.10, and 5.63 m/s$^2$, respectively. Notably, the vertical component exhibits the highest PGA, exceeding those of both horizontal components.

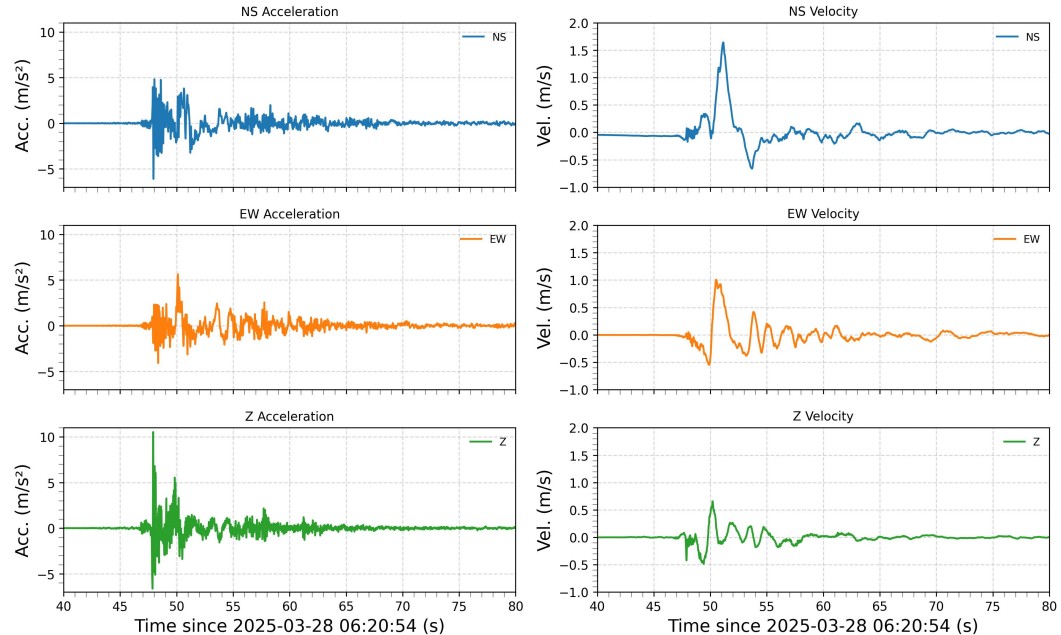

**Figure 6.** Time series of the processed mainshock acceleration and velocity for all three components at station NPW.

– **Acceleration response spectra**. The acceleration response spectra shown in Figure 7 further high-
light the significant vertical accelerations during the mainshock, especially for short-period oscilla-
tors. The short-period spectral amplitudes for the two horizontal components are comparable, with
a pronounced spectral peak observed around 0.15 s (i.e., $\sim 6.5$ Hz), at which the fault-parallel (NS)
component exhibits approximately twice the amplitudes of the fault-normal component (EW). The
horizontal ground motion is compared against a ground motion prediction model (GMPM) appropri-
ate for the region in question, namely Boore et al. (2014). In this case we first define the horizontal
ground motion in terms of RotD50, the median values of the response spectra of the two horizontal
components projected onto all non-redundant azimuths (Boore, 2010), which is shown in the black
line in Figure 7. For comparison with the GMPM, we first calculate the distance to the finite-fault
rupture taken from the USGS Finite Fault service (U.S. Geological Survey, 2025a). The site condi-
tion for the station is assumed as $V_{S30}$ (the time-average shear-wave velocity up to 30 m depth) 360
m/s, while the required basin depth parameters $Z_{1.0}$ (depth to the $V_S =1$ km/s velocity layer) and
$Z_{2.5}$ (depth to the $V_S =2.5$ km/s velocity layer) are predicted from the $V_{S30}$ using empirical relations
proposed by Chiou and Youngs (2014) and Campbell and Bozorgnia (2014), respectively. As seen in

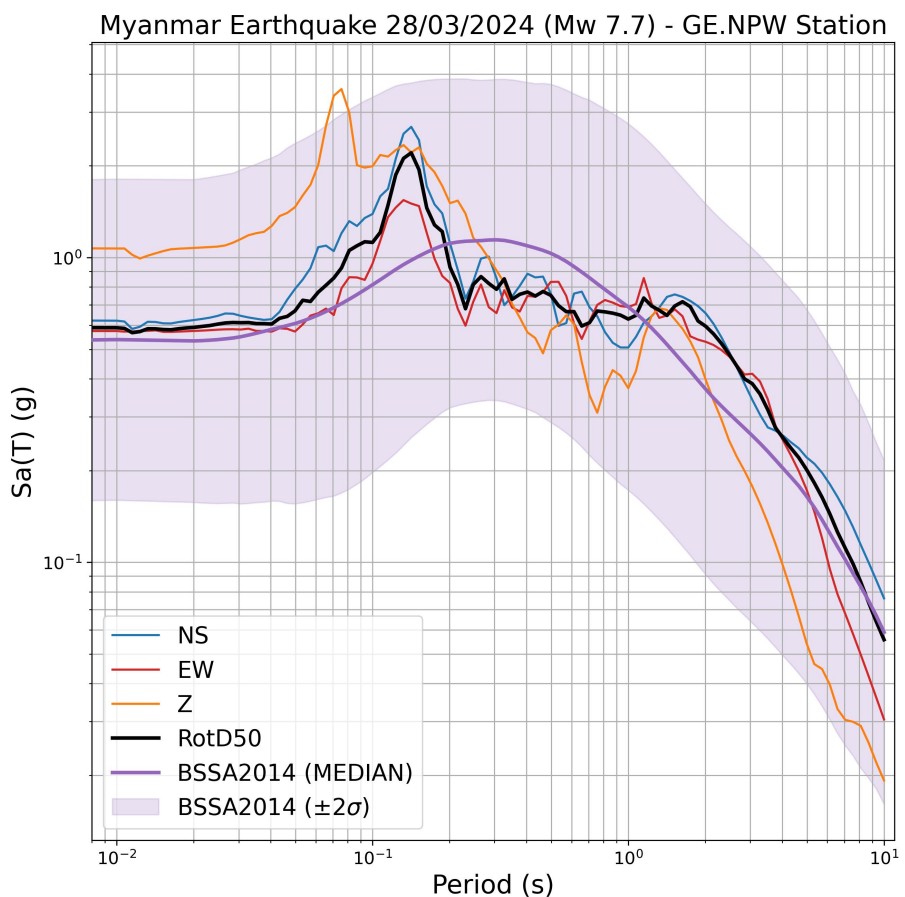

**Figure 7.** Acceleration response spectra for the mainshock recorded at station NPW. Spectra are shown for all three components individually (see legend), as well as for the RotD50 horizontal combination (black), see text. Observed spectra are compared with the median predictions from the Boore et al. (2014) ground motion model.

Figure 7 for these assumed parameters, the observed response spectra for RotD50 fall well within the aleatory variability of the ground motion prediction model across the 0.01 s to 10.0 s period range.

– **Vertical to Horizontal response spectra ratio**. Although vertical-to-horizontal spectral ratios (V/H) greater than 1 are often observed in near-fault recordings of large earthquakes on soft sites, Figure 8 shows that the V/H ratios computed for the mainshock exceed the median predictions by Bozorgnia and Campbell (2016) across all periods. However, the observed ratios generally remain within two standard deviation above the median. A particularly pronounced peak at 0.07 s (approximately 15 Hz) stands out and warrants further investigation.

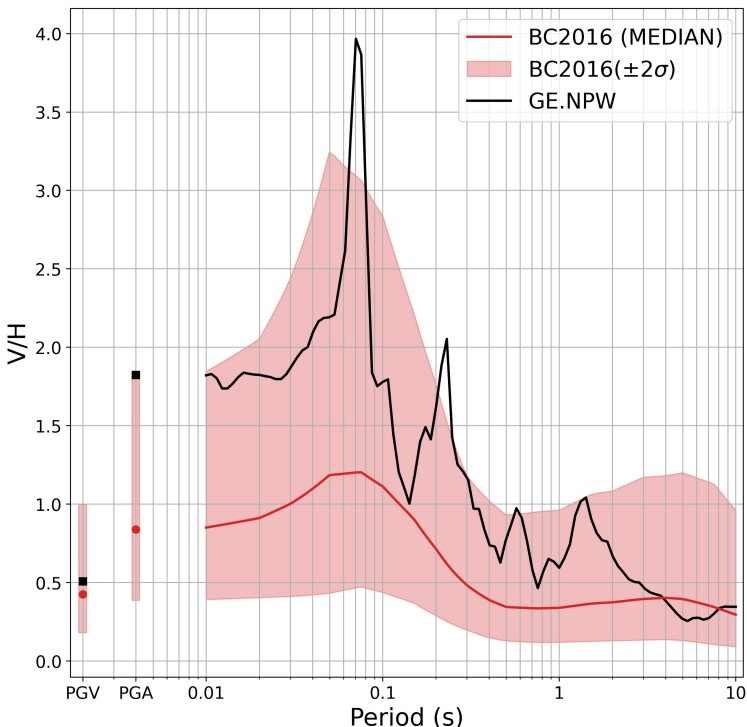

**Figure 8.** Vertical-to-horizontal (V/H) response spectral ratios as a function of period, considering the RotD50 horizontal combination. The V/H computed for the mainshock (black line) is compared with the median predictions $\pm$ two standard deviations from the Bozorgnia and Campbell (2016) model. Ratios for peak ground acceleration (PGA) and peak ground velocity (PGV) are also shown.

– **Fault-normal velocity pulse**. Figure 9 shows the amplitude of the Stockwell transform (Stockwell et al., 1996) for the EW velocity component of the mainshock. The dominant frequency associated with the main velocity pulse is approximately 0.3 Hz, followed by higher-frequency shaking centered around 0.8 Hz. A comparison with previously published observations, shown in Figure 10, indicates that the pulse period is consistent with the general trends reported for other earthquakes (Shahi and Baker, 2014; Türker et al., 2023; Yen et al., 2021, 2024), although it lies near the lower end of the observed range. The temporal evolution of the instantaneous frequency (IF), shown in Figure 9 and defined as the time derivative of the complex phase, provides valuable insight into the non-stationary characteristics of the ground motion. At the onset of strong motion, the IF exhibits a sharp increase, marking the transition from background noise to significant energy input. During the large velocity pulse, the IF decreases from an early peak consistent with the dominance of lower

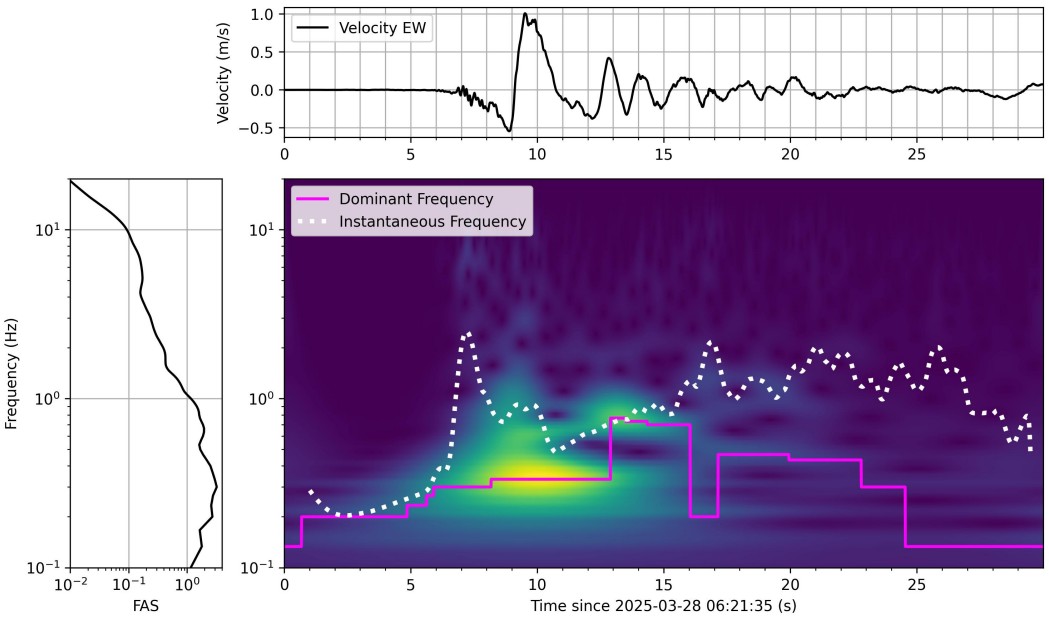

**Figure 9.** Stockwell transform analysis of the velocity pulse (fault normal component), see text. The marginal plot on top is the time series; the marginal plot on the left is the Fourier amplitude spectrum (integral over time for each frequency of the Stockwell transform).

frequencies during this interval. After the pulse, the IF shows increases gradually, suggesting the presence of higher-frequency energy in the later part of the signal.

– **Comparison of engineering seismology parameters with the 2023, Türkiye earthquake**. When comparing the computed PGAs and peak ground velocities (PGVs), calculated as ROTD50, with those recorded during the 2023 Türkiye earthquake (Figure 11), consistent amplitude at similar distances are observed. Figure 11 also shows a comparison with the predictions from the Boore et al. (2014) ground motion model, using the linear site amplification term corresponding to $V_{S30}$=360 m/s.

For both events, the observations fall within the median prediction $\pm$ one standard deviation, with the NPW values close to the median. This good agreement suggests that the actual $V_{S30}$ at station NPW is likely close to or lower than the assumed 360 m/s, which is consistent with the topography-based estimate of 302 m/s (Wald and Allen, 2007).

    Finally, since station NPW has been continuously recording since the onset of the seismic sequence, nu-

245 merous high-quality aftershock recordings are now available in the archive. Figure 1 presents the EMSC (European-Mediterranean Seismological Centre) locations of the mainshock and 61 aftershocks with mag-

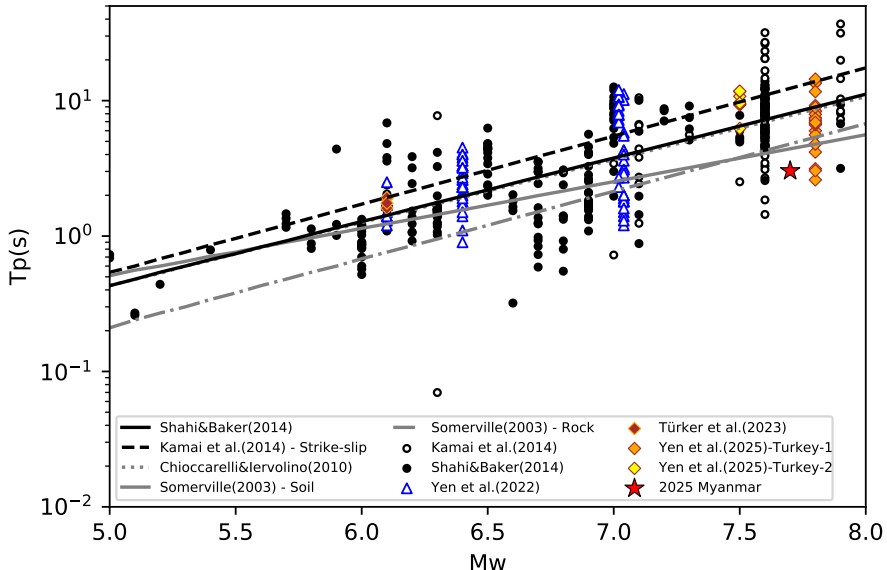

**Figure 10.** Comparison of the pulse period, $Tp$ of the Myanmar earthquake with literature values (compiled by Yen et al. (2024)), as a function of earthquake moment magnitude. Red stars represent the strongest pulse of 2025 Myanmar earthquake. Note that the pulse periods published by Yen et al. (2024) have been computed using the Shahi and Baker (2014) method but the pulse period of the Myanmar earthquake has been measured based on the Stockwell transform of the fault normal component, see Figure 9. The lines show the regressions of Shahi and Baker (2014) (black solid line), Chioccarelli and Iervolino (2010) (gray dotted line), and Somerville (2003) (gray dashed-dotted-dashed line). Black dots represent the pulses identified from the NGA-West2 database (Ancheta et al., 2014) in the study of Shahi and Baker (2014). Open circles represent the fling-step pulses published by Kamai et al. (2014). Blue triangles represent the pulses identified in Taiwan, Japan, and New Zealand (Yen et al., 2021). Colored diamonds represent 2023 Türkiye doublet (Yen et al., 2024) and 2022 Düzce earthquake (Türker et al., 2023). Figure modified and adapted from Yen et al. (2024).

nitudes greater than 3.0, recorded up to April 7, 2025. Among these, Figure 12 also shows the processed acceleration and velocity time histories of a well-recorded $M_w$ 6.7 aftershock that occurred just 12 minutes after the mainshock, at 06:20:54 UTC, approximately 220 km from station NPW. Another example

is the $m_b$ 5.1 aftershock recorded on March 29, 2025, at 09:20:47 UTC, located only 12.3 km from the station. These events highlight the value of the NPW station for capturing both near-fault and far-field ground motions throughout the sequence. The 61 events are part of the data set used to investigate the site resonances following the approach by Lai et al. (2025).

– **Site resonance frequencies**. Figure 13 shows the broadband resonance curves at station NPW, com-
255 puted using the time-frequency resonance analysis method adapted from Lai et al. (2025). We se-

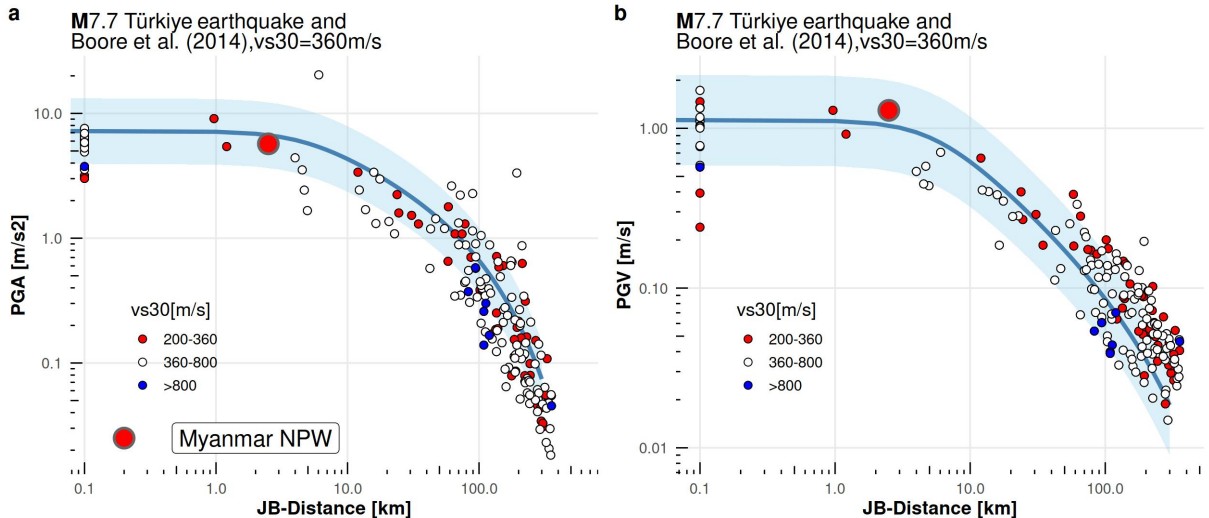

**Figure 11.** Comparison of peak ground acceleration (PGA, left) and peak ground velocity (PGV, right) recorded at station NPW during the $M_w$ 7.7 Myanmar mainshock (large red circle) with the median predictions (solid line) and $\pm 1$ standard deviation bounds (shaded area) of the Boore et al. (2014) ground motion model. For reference, peak values recorded during the $M_w$ 7.7, 2023 Türkiye earthquake (Lanzano et al., 2021) are also shown.

lected aftershock records with PGA ranging from 1 to 10 cm/s$^2$, recorded by accelerometers up to 7 April 2025, resulting in a dataset of 20 events with magnitudes between 3.5 and 6.7. For each event, we applied a time window of 15 s for magnitudes $\leq$ 6.0 and 20 s for magnitudes > 6.0, beginning 20% of the window duration before the PGA time and extending 80% beyond it. The Stockwell transform was then applied to the windowed time series of the derivative of acceleration, acceleration, velocity, and displacement. The squared spectral amplitudes, representing signal energy over time, were normalized between 0 and 1. We then computed three percentiles along the time axis to obtain a smooth representation of energy distribution in the frequency domain, referred to as the resonance curve. The peaks of the curves serve as proxies for the frequencies of linear site amplification but housing effects cannot be ruled out. The station has a predominant resonance frequency around 12-13 Hz. Although this peak is dominant, a secondary peak around 7–8 Hz is also present in the resonance curve, albeit somewhat obscured by the taller primary peak. Additionally, relatively smaller and less prominent peaks are observed at 2.5-3 Hz, 1.4-2 Hz, and approximately 0.9-1.1 Hz.

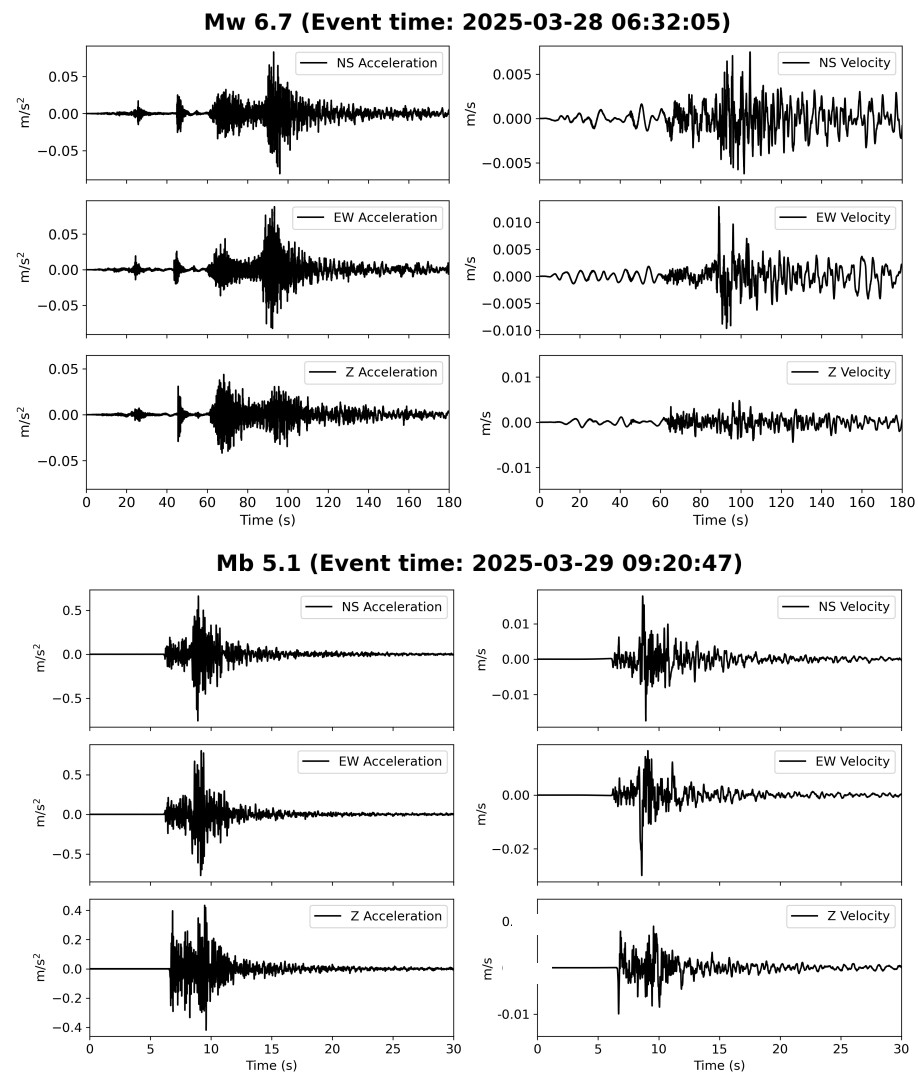

**Figure 12.** The processed acceleration and velocity waveforms of the $M_w$ 6.7 and $m_b$ 5.1 aftershocks, recorded by the accelerometer installed in NPW, are shown. The $m_b$ 5.1 aftershock is located approximately 12.3 km from station NPW (Figure 1).

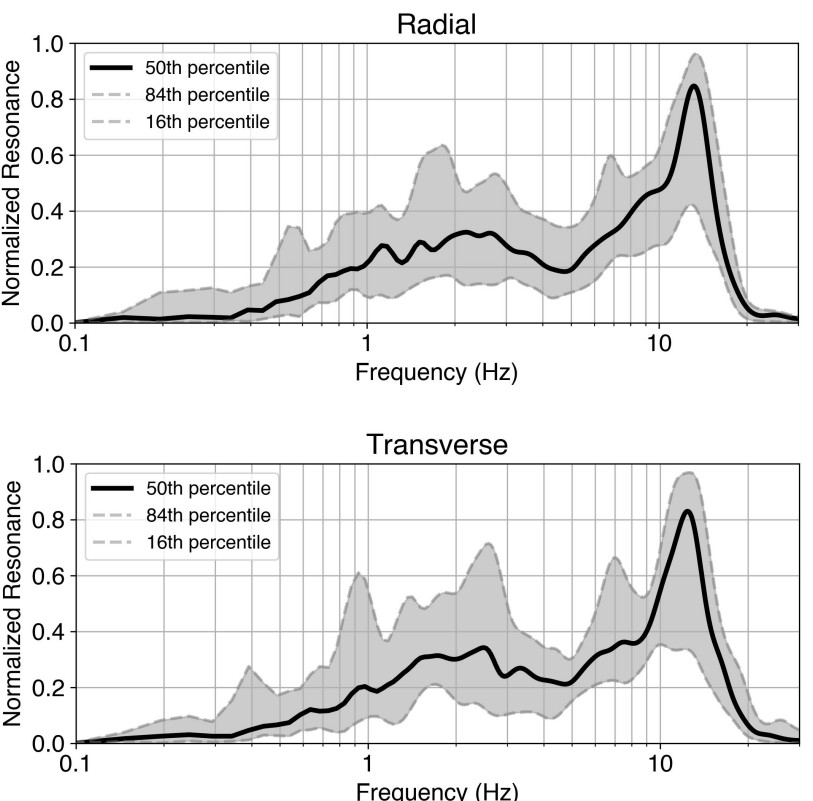

**Figure 13.** Resonance curves at station NPW, computed from the radial and transverse components of aftershock records using the time-frequency resonance analysis method adapted from Lai et al. (2025), see text for further details.

## 5 Code and data availability

— Software and processed data are available at Zenodo under https://doi.org/10.5281/zenodo.15921214 (Bindi et al., 2025).

— Waveform data and metadata for the GEOFON seismic network (GEOFON Data Centre, 1993) with its station NPW are available via standard FDSN web services and formats (https://www.fdsn.org/services) alongside with other custom services and products. We note that the data downloaded 275 through this service will have no timing correction applied; if you need absolute timing, you need to correct the timing yourself.

— Station metadata, including instrument responses for NPW, are available through the FDSN Station Web Service. StationXML format metadata can be obtained using the following query: https://geof

on.gfz.de/fdsnws/station/1/query?level=response&net=GE&station=NPW. On April 3, 2025, issues
with the timing quality and metadata of all channels at NPW were detected. The metadata have been
updated on April 4, 2025 to address these issues (https://geofon.gfz-potsdam.de/forum/t/metadata
-for-ge-npw/32635/4), and users should ensure they have the most recent metadata when analyzing
data from this station.

– Waveform data from NPW can be retrieved from GEOFON using the FDSN fdsnws-dataselect Web
Service http://fdsn.org/webservices/. To request specific time windows of data, one queries the
service, providing parameters such as network code (net), station code (sta), channel (cha), start
time (starttime), and end time (endtime). For example for a 10 minutes time window around the
mainshock including only the 20 Hz broadband vertical channel: https://geofon.gfz.de/fdsnws/datas
elect/1/query?net=GE&sta=NPW&cha=BHZ&starttime=2025-03-28T06:15:00&endtime=2025-
03-28T06:25:00

Due to instability of the internet connection to Myanmar data gaps are present in the continuous
recordings and efforts are ongoing to back-filling gaps when data are available locally at Myanmar
and bandwidth allows.

**Additional data access tools**

– Live seismogram (browsable daily plots of the broadband vertical component): https://geofon.gfz.d
e/waveform/liveseis.php?station=NPW&date=2025-03-28

– Interactive availability calendar view (2025): https://geofon.gfz.de/waveform/archive/data.php?nco
de=GE&year=2025

– fdsnws-availability (average availability 01.01 – 07.04.2025 90%): https://geofon.gfz.de/fdsnws/ava
ilability/1/query?network=GE&station=NPW&start=2025-01-01T00:00:00&end=2025-04-07T00:
00:00

– 2025 WFcatalog quality metrics: availability, RMS, gaps, overlaps, records, timing quality (2025):
https://geofon.gfz.de/eidaws/wfcatalog/1/query?net=GE&station=NPW&channel=HHZ,HNZ&sta
rt=2025-01-01T00:00:00.000Z&end=2025-04-07T23:59:59.999Z&include=all

### The Power Spectral Density PSD (Figure 3)

The PSD for the period from 01.01.2025 to date for all components at 100 Hz can be retrieved as follows for the vertical strong motion channel:

- https://geofon.gfz.de/eidaws/seedpsd/1/histogram?&cmap=viridis&fontsize=12&grid=true&mode=true&noise=true&percentiles=false&&dpi=600&network=GE&station=NPW&location=*&channel=HNZ&nodata=404&start=2025-01-01&end=2025-12-31

Replace HNZ in above by HNN, HNE for the horizontal strong motion, and by HHZ, HHN, HHE for the broadband PSDs. The channel naming follows the SEED standard (https://www.fdsn.org/pdf/SEEDManual_V2.4_Appendix-A.pdf).

**Additional data used**

- Waveform data from stations the national network of Myanmar (Department of Meteorology and Hydrology - National Earthquake Data Center, 2016, MM) and Thailand (TM network, https://earthquake.tmd.go.th/) were used for ambient noise analysis to constrain the clock drift of NPW.

- In Figure 11, the peak parameters for the 2023 Türkiye earthquake are relevant to networks KO (Kandilli Observatory and Earthquake Research Institute, Boğaziçi University, 1971) and TK (Disaster and Emergency Management Authority, 1973), originally downloaded from https://tdvms.afad.gov.tr/

**Additional software used**

- Computations are done in Python (Van Rossum and Drake, 2009) and R (R Core Team, 2024), using ObsPy (Beyreuther et al., 2010)

- The stream2segment software used to download and process the waveform is available by Zaccarelli (2018). Templates for the data download and process are provided with the distribution.

- The eGSIM software used to compare ground motion model predictions and observation is available by Zaccarelli and Weatherill (2020).

## 6 Concluding remarks

The installation of station NPW was made possible through a collaborative effort between the Department of Meteorology and Hydrology (DMH) in Napyitaw and the GFZ Helmholtz Centre for Geosciences. Similar to other initiatives like the upgrade of the MM network as part of an effort supported by the USGS and USAID(Thiam et al., 2017), this collaboration underscores the critical role of capacity building in supporting the development of high-quality, open-access seismic data that can serve both national 335 monitoring efforts and the global scientific community. By strengthening local expertise, infrastructure, and collaborative networks, such initiatives help establish and sustain robust seismic monitoring systems that provide valuable observations for advancing earthquake science and hazard assessment. In the case of the $M_w$ 7.7 Myanmar earthquake sequence in 2025, NPW is the only station to provide near-fault, on-scale recordings of the mainshock and the largest aftershocks. It thus adds a rare example of a near-fault 340 record of a supershear rupture to the global strong motion databases. As such, the NPW near-fault data set presented in this study not only support fundamental research aimed at improving our understanding of earthquake processes and seismic hazard, but also informs practical strategies for risk mitigation, urban planning, and resilient infrastructure design in vulnerable regions.

*Author contributions.* All authors provided contributions to the conceptualization and finalization of the article, coordinated by DB and FT.
Regarding the figures: MHY prepared Figures 1 and 10; AS prepared Figures 2 and 3; CSS prepared Figure 4; JS prepared Figure 5; DB prepared Figures 6, 9 and 11; GW prepared Figures 7 and 8; STL prepared Figures 12 and 13. The 2016 training course was organized and managed by CM.

*Competing interests.* The authors declare no competing interests.

*Acknowledgements.* The 2016 International training course on Seismology, Seismic data analysis, Hazard assessment and Risk mitigation 350 (Naypyitaw, 26 September-21 October, 2016) had been organized and sponsored by the GFZ Helmholtz Centre for Geosciences (GFZ) and the Department of Meteorology and Hydrology in Naypyitaw (Myanmar), and co-sponsored by the German Federal Foreign Office (FFO), Berlin, Germany. We thank D. Kroll for her support in the organization of the training. We thank M. Motagh and the Remote Sensing for Geohazards group (GFZ) for sharing Figure 2a. The article processing charges for this open-access publication were covered by the GFZ.

Finally, we appreciated the valuable feedback and suggestions provided by Brad Aagaard, Susan Hough, one anonymous reviewer, and by the topic editor Andrea Rovida.

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
