# Peer review of "Capacity Building Enables Unique Near-Fault Observations of the destructive 2025 $M_w$ 7.7 Myanmar Earthquake"

_Earth System Science Data, 2025_

## Author Response (AR1)

**Reviewer 1**

Useful and timely paper, good to publish as is. I only have one technical note, that is, the referece to Luzi et al. (2020) in Figure 10 is not correct in my opinion. Please note:

- The reference to the ESM is the paper by Lanzano et al. (SRL 2021) [doi: https://doi.org/10.1785/0220200398];

- The data are those of the seismic networks TK (https://doi.org/10.7914/SN/TK) and KO (https://doi.org/10.7914/SN/KO), to be cited;

- The data were originally donloaded from https://tdvms.afad.gov.tr/, to be mentioned/ acknowledged.

We thank the Reviewer for the suggestions provided. We made the following changes:

- In the caption of Figure 11 (which corresponds to Figure 10 of the original submission), we updated the ESM reference to Lanzano et al. (2021);

- In the **Additional Data Used** section, we added the following: In Figure 11, the peak parameters for the 2023 Türkiye earthquake are relevant to networks KO (Kandilli Observatory and Earthquake Research Institute, Boˇgaziçi University, 1971) and TK (Disaster and Emergency Management Authority, 1973), originally downloaded from https://tdvms.afad320.gov.tr/

We also updated the **References.**
* * *
**Reviewer 2**

We thank Brad Aagaard for his detailed comments and suggestions. Below, we provide answers to each of his questions.

**Abstract**
Mention network (and network code) in addition to station code in first sentence.
We added the network name and code

What is meant by "local station"? I suggest rephrasing in terms of distance from the fault rupture.
We changed "local station" to "the only station providing near-fault, on-scale measurements of "

**Introduction**
Line 16: Provide a reference for moment magnitude.
The moment magnitude is taken from the Geofon event catalog (Quinteros et al., 2021): https://geofon.gfz-potsdam.de/fdsnws/event/1/query?eventid=gfz2025gbpv&format=text

Lines 17-18: Provide a reference for the shaking intensity and number of fatalities.
We added the references to both ShakeMap and PAGER services:

U.S. Geological Survey, 2025, ShakeMap, version 2025-06-06 13:36:23 (UTC) at URL https://earthquake.usgs.gov/earthquakes/eventpage/us7000pn9s/shakemap/intensity
U.S. Geological Survey, 2025, PAGER, version 2025-04-16 05:22:25 (UTC) at URLhttps://earthquake.usgs.gov/earthquakes/eventpage/us7000pn9s/pager

Line 23: Please elaborate and quantify the "sparse" seismic instrumentation. For example, what is the average station density? What is the distribution between strong-motion and broadband instruments?
We added a new Figure 1 with a map showing also the location of the MM stations operating in the region. A detailed description of the MM network is available in Thiam et al (2017), already included in the reference list. MM stations are equipped with both broad band and accelerometric sensors.

Information about the MM network can be retrieved from IRISDMC: http://service.iris.edu/irisws/fedcatalog/1/query?
net=MM&format=text&includeoverlaps=true&nodata=404

Line 26: Provide FDSN network code for GEOFON and the URL of the data center. This information is included later, but it is helpful to include it early on in the manuscript.
We added the network code; the URL is already given in the network's reference (see GEOFON Data Centre, 1993).

Line 27: How many other stations are in the vicinity (within 100 km of the rupture)? I assume these are all broadband sensors and were clipped.
GE.NPW is the only station providing on-scale, near-fault records for the mainshock. Among the MM stations, MDY, TKU, SIM, KTA, and HKA have no data available for the mainshock. Station NGU is located at a Joyner-Boore distance (Rjb) of approximately 114 km, and has on-scale accelerometric records (HN channels) and clipped broadband records (HH channels). Station YGN (Rjb =137 km) has on-scale HN and clipped HH records. Station TGI has no usable HN records and clipped HH channels; station KTN (Rjb=367 km) has on-scale HN records and clipped HH ones. Station CHTO, which is part of the IU network and is located in Thailand (Rjb=268 km), provides additional accelerometric data. Locations of the MM stations are shown in the new Figure 1.

Further strong motion information available at:
https://www.strongmotioncenter.org/cgi-bin/CESMD/iqr_dist_DM2.pl?
IQRID=us7000pn9s&SFlag=0&Flag=2

**Background collaboration**
Line 62: Why was NPW created as part of the GE network and not integrated into the MM network? Were any other stations installed that are part of the GE network and not the MM network?
Affiliation to the GE (GEOFON) global seismic network provides a long-term collaborative framework in which partners commit to open data sharing. It facilitates regional and global interaction, supporting local monitoring and contributing to tsunami early warning systems, which are also relevant for Myanmar within the wider Southeast Asian context. Despite the complex circumstances, this framework has ensured the continued operation of the station as explained in the text.
The GE network tag serves mainly as an administrative label that distinguishes GEOFON policies from those of regional and local networks, which may differ. The partnership is formalized through an agreement in which GFZ commits to providing sustainable support and open data sharing under a CC BY license. As co-owners of the data, partners are still free to distribute data from their facilities according to their preferred policies.
Under the GE affiliation, GFZ provides long-term support, including hardware maintenance as needed, remote assistance, SeisComP training and integration. This streamlined approach eliminates administrative barriers, enabling DMH to integrate all available stations, including the MM and other regional open stations/networks, from the start. In 2016, DMH specifically required a unified framework for data acquisition and processing. Over the years, the collaboration has included not

only mutual support to operate the NPW station, but also to maintain and update the SeisComP system (including additional on-site and in-Germany training).

Figure 1: For panel (a), please explain the colors (add color bar) and how the coherence image is associated with the photograph.
The previous Figure 1 is now Figure 2. We added the color bar and updated the captions as: a) location of station NPW with respect to the main rupture plane of the 2025, Mw 7.7 Myanmar earthquake as depicted by pixel offset tracking \citep{Strozzi02} on Sentinel 1 (\url{https://sentiwiki.copernicus.eu/web/s1-mission}) and Sentinel 2 (\url{https://sentiwiki.copernicus.eu/web/sentinel-2}) data (courtesy of M. Motagh and the Remote Sensing for Geohazards group, GFZ Helmholtz Centre for Geosciences; see \citet{Vera25preprint} for details; Map data \textsuperscript{\textcopyright}2025 Google).

Strozzi, T., Luckman, A., Murray, T., U., W., & Werner, C. L. (2002). Glacier motion estimation using SAR offset-tracking procedures. IEEE Transactions on Geoscience and Remote Sensing, 40 (11), 2384-2391. doi: https://doi.org/10.1109/TGRS.2002.805079

Vera, F. Carrillo-Ponce A., Crosetto S., et al. Supershear Rupture Along the Sagaing Fault Superhighway: The 2025 Myanmar Earthquake. ESS Open Archive . June 19, 2025. DOI: 10.22541/essoar.175034871.19414276/v1

**Instrumental settings**
Line 96: Please quantify "reasonable noise levels".
The PDFs' mode falls within the lower and upper limits of the New Noise Model proposed by Peterson (1993). In particular, it is less than 10-20 dB from the lower limit for periods longer than one second. The updated text reads: "The mode of the power spectral density (Figure 3) falls within the lower and upper limits of the New Noise Model proposed by \citet{Peterson93}. In particular, it is less than 10–20 dB from the lower limit for periods longer than 1 second."

Peterson J. (1993). Observations and modeling of seismic background noise, U.S. Geol. Surv. Open-File Rept. 93-322 , U.S. Department of the Interior, U.S. Geological Survey

Line 98: Please rephrase to clarify what is meant by "conditions in which the station operates".
We have revised the text. The phrase "conditions in which the station operates" has been rephrased to : "......, although NPW is not installed in a remote and quiet area, ……".

Line 101: Please elaborate on what adaptations were necessary or clarify the adaptations being referred to.
The setup was discussed and agreed upon with the local partners. Instead of the usual GEOFON hardware, which would have been more complicated to operate, it was adapted to their actual capacity and needs by procuring and shipping dedicated hardware with which the partners were already familiar.

Line 104: What factors are limiting data availability?
This is mostly related to unreliable local power supply and repeated hardware failures, which was difficult to replace due to the complex situation. We are currently trying to ship new hardware capable of time synchronisation over the network, which we expect will improve both timing quality and availability.

Figure 2: In panels (a) and (b) indicate the meaning of the blue and green color bars. The annotation is too small to read in panels (c) and (d).

The blue and green bars indicate the available and used segments for the analysis, respectively. The original Figure 2 (now Figure 3) has been updated with larger annotations.

Line 115: What are the distances to these neighboring stations?
We added Figure 1 to provide more detail about the location of the Myanmar Meteorological (MM) network. We also mentioned the temporary networks that were operated in Myanmar in the text. We included relevant data sources and citations to clarify the available seismic data for the region.

Line 122: Consider reorganizing this section so that the discussion of noise level is closer to its first mention.
We moved lines from 122 to 127 (of the original version) to be closer to where the noise levels are mentioned.

Line 133: "In April 2025, immediately following the 28 March 2025 ..." Please replace "immediately" with a more quantitative term. This sentence is confusing as the communication seems to be lost on March 28, but the sentence mentions April 2025.
We have revised the sentence to avoid confusion (Line 146-148): "Following the $M_W$ 7.7 earthquake on March 28, 2025, the contact was initially lost a few seconds after the P onset. The condition of the station was unknown to GEOFON staff at that time. The station started transmitting data again on April 1, 2025, four days after the earthquake. "

Line 146: This statement repeats points made earlier in the manuscript.
Thank you for pointing this out. We have removed the repeated statement.

**Data quality and parameters of engineering interest**
Line 151: This statement repeats points made earlier in the manuscript.
We removed the statement.

Line 160: Please specify which method and parameters were used to estimate the time of the P arrival.
For the task of extracting from the continuous streams the time windows of interest, we computed the theoretical P-arrival time considering the earthquake location and the AK135 (Kennett, Engdahl & Buland, 1995) global velocity model.

Kennett, B.L.N. Engdahl, E.R. and Buland R., (1995). Constraints on seismic velocities in the Earth from travel times, Geophys J Int, 122, 108-124

Line 167: It would be helpful to illustrate the baseline corrections applied in the double integration. Baseline correction that preserves static offsets is an active research area, especially as the number of records with static offsets increases. It would be helpful to include a brief discussion of the methods considered and why the chosen method was selected. Were there deficiencies in the other methods?
The applied procedure is summarized in the manuscript. We added to the Zenodo repository (https://doi.org/10.5281/zenodo.15921214) the python code used to perform the double integration.

Line 178: How were the corner frequencies in the bandpass filtering chosen?
We follow Puglia et al (2018), with small adjustments after visual inspection of the results. The reference has been added.

Puglia, R., Russo, E., Luzi, L. et al. Strong-motion processing service: a tool to access and analyse earthquakes strong-motion waveforms. Bull Earthquake Eng 16, 2641–2651 (2018). https://doi.org/10.1007/s10518-017-0299-z

We confirm that the results shown are acceleration response spectra computed with 5% damping (information provided in the text).

Following the suggestion from Dave Boore, we prefer to keep the acronym GMPMs (Ground motion prediction models): https://daveboore.com/daves_notes/Thoughts%20on%20the%20acronyms%20GMPE,%20GMPM,%20and%20GMM.v2.pdf

We changed the original Figure 6 by showing the results for the BSSA14 model.

Yes, it is possible that some of the peaks shown in the response spectra are related to housing effects or resonances of nearby structures. In particular, the narrow peak at approximately 15 Hz on the vertical component may be related to the power supply generator. These features will be the subject of future investigations.

We updated the reference for the finite fault model as follows:
U.S. Geological Survey, 2025, Finite Fault, version 22025-04-02 14:19:57 (UTC) at URL https://earthquake.usgs.gov/earthquakes/eventpage/us7000pn9s/finite-fault

Since no site characterization is available and given the local geological conditions, we applied the predictive models assuming a Vs30 value of 360 m/s (i.e., the boundary between the C and D classes in the NEHRP classification). Using the topographic gradient as a proxy for Vs30 (Wald and Allen, 2007) yielded an inferred value of 302 m/s, supporting this assumption. We have added a reference to Wald and Allen (2007).

Thank you for the suggestion. Highlighting supershear events using a consistent symbol/color could indeed help clarify potential patterns in the data. We agree that this would be a valuable addition to future work.

We rephrased the text by indicating that "For both events, the observations fall within the median prediction $\pm$ one standard deviation, with the NPW values close to the median".

The mention of the topography-based estimate of Vs30 has been moved to Line 200 (now Line 240), and the citation has been added.

Thank you for the comment. To improve clarity, we have separated the right panels of the original Figure 11 into a new, independent Figure 12 (and we used the map to prepare the new Figure 1). Additionally, we enlarged the figure's annotations for better readability.

Line 254: Is the peak in the V/H ratio shown in Figure 7 consistent with the resonance?
Figure 7 (now Figure 8) is relevant to the mainshock recording whereas Figure 12 (now Figure 13) shows the results for the rotated horizontal components considering smaller events. The mainshock spectra show peculiar peaks (e.g., the narrow peak at approximately 14 Hz on the vertical) which are not observed for other events, or when analysing the power spectral densities (PSD) of noise windows before the sequence. Future investigations will address the stationarity of the spectral features and whether they are due to site amplification, housing effects, or anthropogenic noise.

**Code and data availability**
Line 270: Indicate the date(s) when the metadata were updated.
On April 4, 2025, https://geofon.gfz.de/forum/t/metadata-for-ge-npw/32635/4

**Zenodo archive**
Add README.txt with a description of each of the files.
Add units to the headers in the time history and spectral acceleration files.
Consider combining the time history files so that all three components are in a single file, and name the file in a way that is self-explanatory when downloaded (for example, include the event name, station, and acceleration).
The purpose of the Zenodo repository is simply to show an easy way to download data from NPW and apply a basic processing. Users can modify the script as needed. We have added to the repository the python code used to perform the double integration preserving the static offset.